

# Occupancy of wild southern pig-tailed macaques in intact and degraded forests in Peninsular Malaysia

Anna Holzner[1,2,3], D. Mark Rayan[4,5], Jonathan Moore[6,7],
Cedric Kai Wei Tan[8,9], Laura Clart[2], Lars Kulik[1], Hjalmar Kühl[10],
Nadine Ruppert[3] and Anja Widdig[1,2,10]

[1] Department of Human Behaviour, Ecology and Culture, Max Planck Institute for Evolutionary Anthropology, Leipzig, Germany
[2] Behavioural Ecology Research Group, Institute of Biology, University of Leipzig, Leipzig, Germany
[3] School of Biological Sciences, Universiti Sains Malaysia, Pulau Pinang, Malaysia
[4] Durrell Institute of Conservation and Ecology (DICE), University of Kent, Canterbury, United Kingdom
[5] Wildlife Conservation Society (WCS) Malaysia Program, Petaling Jaya, Malaysia
[6] School of Environmental Science and Engineering, Southern University of Science and Technology, Shenzhen, China
[7] School of Environmental Sciences, University of East Anglia, Norwich, United Kingdom
[8] Wildlife Conservation Research Unit, Department of Zoology, University of Oxford, Oxford, United Kingdom
[9] School of Environmental and Geographical Sciences, University of Nottingham Malaysia, Semenyih, Malaysia
[10] German Centre for Integrative Biodiversity Research (iDiv), Halle-Jena-Leipzig, Germany

Corresponding author
Nadine Ruppert, n.ruppert@usm.my

## ABSTRACT

Deforestation is a major threat to terrestrial tropical ecosystems, particularly in Southeast Asia where human activities have dramatic consequences for the survival of many species. However, responses of species to anthropogenic impact are highly variable. In order to establish effective conservation strategies, it is critical to determine a species' ability to persist in degraded habitats. Here, we used camera trapping data to provide the first insights into the temporal and spatial distribution of southern pig-tailed macaques (*Macaca nemestrina*, listed as 'Vulnerable' by the IUCN) across intact and degraded forest habitats in Peninsular Malaysia, with a particular focus on the effects of clear-cutting and selective logging on macaque occupancy. Specifically, we found a 10% decline in macaque site occupancy in the highly degraded Pasoh Forest Reserve from 2013 to 2017. This may be strongly linked to the macaques' sensitivity to intensive disturbance through clear-cutting, which significantly increased the probability that *M. nemestrina* became locally extinct at a previously occupied site. However, we found no clear relationship between moderate disturbance, *i.e.*, selective logging, and the macaques' local extinction probability or site occupancy in the Pasoh Forest Reserve and Belum-Temengor Forest Complex. Further, an identical age and sex structure of macaques in selectively logged and completely undisturbed habitat types within the Belum-Temengor Forest Complex indicated that the macaques did not show increased mortality or declining birth rates when exposed to selective logging. Overall, this suggests that low to moderately disturbed forests may still constitute valuable habitats that support viable populations of *M. nemestrina*, and thus need to

be protected against further degradation. Our results emphasize the significance of population monitoring through camera trapping for understanding the ability of threatened species to cope with anthropogenic disturbance. This can inform species management plans and facilitate the development of effective conservation measures to protect biodiversity.

# INTRODUCTION

Tropical rainforests are highly complex ecosystems that exhibit exceptional biodiversity (*Gallery, 2014*). Yet, continuing human population growth, expanding infrastructure and the intensive cultivation of crops lead to dramatically increasing deforestation rates, which are the major threat to these remarkable habitats (*Rosa et al., 2016*). Southeast Asia, for example, one of the world's most biodiverse regions and home to many charismatic primate species (*Myers et al., 2000*; *Sodhi et al., 2010*), has lost about 80 million hectares of forest between 2005 and 2015 (*Estoque et al., 2019*).

It is well known that human land use drastically reduces biodiversity and important ecosystem functions of primary forests (*Marques et al., 2019*; *Alroy, 2017*; *Barnes et al., 2014*). Degraded habitats, such as forest fragments, monocultures or urban environments, were shown to include on average 41% fewer species than undisturbed forests (*Alroy, 2017*). Numerous studies have highlighted the negative impact of forest clear-cutting on biodiversity, with the conversion of tropical forests into oil palm plantations being the main driver of deforestation in many Southeast Asian countries (*Koh & Wilcove, 2007*; *Fitzherbert et al., 2008*). The establishment of oil palm monocultures does not only imply severe losses in species richness but also substantially contributes to habitat fragmentation and environmental pollution through chemical fertilizers or pesticide runoffs (reviewed in *Fitzherbert et al., 2008*). Particularly, larger mammals and specialized bird species that have a narrow dietary spectrum may have difficulties in permanently adapting to and surviving in these monocultural landscapes (*Danielsen & Heegaard, 1995*; *Fitzherbert et al., 2008*). The negative effects of habitat degradation on animal populations may further be reinforced by the increase of so-called 'edge effects', describing ecological alterations that result from the development of abrupt, artificial edges of forest fragments (*Didham et al., 1998*). Forest edges open the canopy and dry out the wood, which increases the susceptibility of forests to fire (*Cochrane, 2003*).

Selective logging is one of the most widespread, albeit less intensive, forms of habitat degradation (*Asner et al., 2005*). It refers to the removal of a limited number of economically valuable tree species of a given age in a particular area and/or during a distinct logging cycle (*Johns, 1985*). However, it also implies secondary threats, such as an increased hunting pressure from local communities, as logging roads facilitate human access to forests (*Robinson, Redford & Bennett, 1999*; *Milner-Gulland & Bennett, 2003*). Previous research suggested that selective timber extraction may have less severe effects on

species diversity and abundance than more intensive forms of land use change, *e.g.*, through clear-cutting (*Ibarra et al., 2017*; *Gibson et al., 2011*). However, *Tobias (2015)* highlighted the high variability in the reaction of different species to habitat degradation, with both logging practices as well as species traits, such as diet and body mass, being important factors in determining the effects of selective timber harvesting on wildlife abundance. This is in line with other studies suggesting that generalist feeding tendencies in particular are indicative of the ability of a species to persist in selectively logged habitats (*Vetter et al., 2011*; *Burivalova et al., 2015*). Indeed, some species were found to have a higher abundance in moderately disturbed compared to primary forest habitats, including several ungulates (*Brodie, Giordano & Ambu, 2015*), rodents, and granivorous bird species (*Bicknell & Peres, 2010*). Carnivores, as well as frugivorous forest specialists, on the other hand, were often reported to be confined to undisturbed primary forests (*Brodie, Giordano & Ambu, 2015*; *Tobias, 2015*). In this context, caution is needed when inferring the adaptive capacity of one species from the response of another.

Malaysia is a biodiversity hotspot with high primate diversity but much of its primary forest is being converted into new oil palm plantations, quarries and urban areas (*Vijay et al., 2016*; *Omran & Schwarz-Herion, 2020*). During the past decade, the country has lost 11.3% of its primary forest and 16.8% of tree cover (*Mongabay, 2021*). As reported by the International Union for Conservation of Nature (*IUCN, 2020*), more than one fourth of Malaysia's mammals are threatened with extinction. Among them are 25 non-human primates (hereafter 'primates'; *Roos et al., 2014*), one of which is the southern pig-tailed macaque (*Macaca nemestrina*), a little-studied, predominantly terrestrial species native to the tropical rainforests of Malaysia, Indonesia and southern Thailand (*Ang et al., 2020*). The macaques' diet consists primarily of fruits (*Caldecott, 1986*), indicating their value as seed dispersers (*Ruppert, Mansor & Anuar, 2014*) and, consequently, their potential role in forest regeneration (*Albert et al., 2014*). However, it also suggests that they are frugivorous forest specialists with limited ecological flexibility. The dramatic decline of primary forest habitat in the primates' range, human hunting of macaques for food and the pet trade, and their widespread perception as crop pests have contributed to rapidly decreasing populations during the past few decades (*Linkie et al., 2007*; *Ang et al., 2020*). Only recently, *Ang et al. (2020)* confirmed the macaques' negative population trend and its current status as 'Vulnerable' (*IUCN, 2020*). *Meijaard et al. (2007)* reported a generally high sensitivity of this species to logging, yet the macaques' response to human disturbance remains poorly understood (*Ang et al., 2020*). To date, we lack detailed knowledge on *M. nemestrina*'s distribution, their abundance as well as their ability to cope in anthropogenically impacted habitats. However, these issues are crucial to understand in order to establish effective protection measures ensuring the long-term survival of this and other threatened wildlife species affected by human activities.

Using available camera trapping data, we provide the first insights into the impact of forest degradation through tree felling on the occupancy of *M. nemestrina* in Peninsular Malaysia. In order to obtain a broad picture of the effects of human activities on this species, the study comprised two sites that are characterized by different degrees of human disturbance. Firstly, we used a dynamic occupancy modelling approach

(*MacKenzie et al., 2003*) to assess temporal changes in the macaques' distribution as well as factors potentially impacting dynamics in site occupancy in the highly disturbed Pasoh Forest Reserve (PFR), which was affected by partial clear-cutting and selective logging from 2013 to 2017. Secondly, we investigated the macaques' spatial distribution within the Belum-Temengor Forest Complex (BTFC) from 2011 to 2013, focusing on differences between the undisturbed, strictly protected Royal Belum State Park and the selectively logged Temengor Forest Reserve. This direct comparison enables a better understanding of the immediate effects of selective timber extraction on the site occupancy of *M. nemestrina*. Finally, we assessed potential differences in the macaques' age and sex structure in undisturbed as well as selectively logged forests within BTFC. This can inform about vital parameters of population dynamics, particularly breeding success and survival, and therefore be indicative of population health.

Although *M. nemestrina* readily leaves its natural forest habitat to enter oil palm plantations in search of food (*Ruppert et al., 2018*; *Holzner et al., 2019*), it is described as a shy and elusive macaque species that tends to avoid human-dominated areas (*Bernstein, 1967*; *Oi, 1990*). Recent studies highlighted the macaques' dependency on primary forest habitat as a safe retreat to sleep and socialize (*Ruppert, Mansor & Anuar, 2014*; *Holzner et al., 2021*), confirming previous doubts on their ability to permanently persist in highly disturbed habitats (*Caldecott, 1986*). Accordingly, we predicted that forest clearance negatively affects the macaques' ability to occupy a specific habitat, and hence a general decline in site occupancy in PFR during the sampling period. As a predominantly frugivorous species, *M. nemestrina* may also be sensitive to less intensive forms of habitat degradation, such as selective timber harvesting. Thus, we predicted macaques site occupancy to be lower in selectively logged compared to undisturbed forests within BTFC. In line with this, we predicted measures describing the accessibility of a site to humans, such as the distance to the nearest human settlement or the forest edge, to negatively affect macaque site occupancy. These may serve as a proxy for hunting pressure, which has been suggested to be one of the major risks to wildlife in disturbed habitats (*Milner-Gulland & Bennett, 2003*; *Tilker et al., 2019*). In addition, environmental factors may influence macaque site occupancy. Based on previous studies (*Yanuar et al., 2009*; *McCain & Grytnes, 2010*; *Ang et al., 2020*), we predicted elevation, defining different floristic zones and thus food availability, to be an indicator of the suitability of a site for *M. nemestrina*. *Yanuar et al. (2009)* reported that this species is best adapted to lowland and hill dipterocarp forests up to 900 m above sea level. Importantly, anthropogenic impact on animals' natural habitats may further be associated with demographic changes in wildlife populations (*Klass, Belle & Estrada, 2020*; *Shil, Biswas & Kumara, 2020*). Particularly, a proportionally low number of juveniles resulting from low birth rates can be indicative for a declining population (*Rudran & Fernandez-Duque, 2003*; *Shil, Biswas & Kumara, 2020*). Moreover, skewed adult sex ratios have previously been linked to increased mortality within the dispersing sex in animal populations, owing to the risk of migration (*Zunino et al., 2007*; *Klass, Belle & Estrada, 2020*). Accordingly, we finally predicted differences in the macaques' age and sex structure between habitats with varying degrees of disturbance. Specifically, we hypothesized the ratio of immature to adult

individuals to be lower and the adult sex ratio to be less balanced, potentially with a surplus of females, in disturbed, *i.e.*, selectively logged, forests compared to undisturbed primary forests within the BTFC. As is true for most Cercopithecine primates, female southern pig-tailed macaques are philopatric and form the core of social groups, while male individuals leave their natal group at sexual maturity to breed elsewhere (*Cords, 2012*).

## METHODS

### Study sites

Our study was conducted at two different sites in Peninsular Malaysia, which are characterized by different degrees of human impact. One site is located within the highly disturbed Pasoh Forest Reserve (PFR) in the state of Negeri Sembilan (102°31′0″E, 2°98′0″N). Large parts of the 140 km$^2$ sized PFR were logged between the 1950s and 1970s, today comprising regenerating lowland forest. Only its 4 km$^2$ core area still consists of virgin primary forest (*Fletcher et al., 2012*). Also today, PFR is subject to clear-cutting and selective logging. It is surrounded by oil palm plantations. The second study site is located within the Belum-Temengor Forest Complex (BTFC) in the state of Perak (101°15′0″-101°46′0″E, 5°55′0″-5°0′0″N). BTFC is less intensively impacted by human activities than PFR. With a size of approximately 3,000 km$^2$, it forms part of the second-largest contiguous forest complex in Peninsular Malaysia, comprising lowland, hill and upper dipterocarp, as well as montane forest (*Rayan & Linkie, 2016*).
The Gerik-Jeli Highway divides the forest complex into two areas, *i.e.*, the strictly protected Royal Belum State Park (hereafter 'Belum') in the north, and the Temengor Forest Reserve (hereafter 'Temengor'), where selective logging has been ongoing since the 1970s, in the south (*Rayan & Linkie, 2016*).

### Camera trap setup

This study is based on camera trap data originally collected to assess habitat use of mainland clouded leopards (*Neofelis nebulosa*) in PFR (*Tan et al., 2017*) as well as density and habitat use of Malayan tigers (*Panthera tigris jacksoni*), occupancy of ungulates and interactions between large carnivores in BTFC (*Rayan & Linkie, 2015*, *2016*, *2020*).
All necessary permits and support letters from the Perak State Parks Corporation, the Department of Wildlife and National Parks and the Forestry Department of Perak were acquired prior to data collection. Due to the non-invasive nature of observational studies based on camera trapping, no institutional ethical approval was required.

Detection data of *M. nemestrina* from PFR were provided by the Tropical Ecology Assessment and Monitoring (TEAM) Network (*Fletcher & Campos-Arceiz, 2011*). In PFR, camera trapping was conducted from 2013 to 2017. Each camera trap was active for an average of 32 (standard deviation SD = 5) consecutive days per year. The camera setup covered an area of 120 km$^2$, including a grid of 60 cameras. The spacing between cameras was approximately 1.5 km. Camera images of macaques detected in BTFC were provided by D. Mark Rayan and WWF Malaysia. Here, camera trapping was conducted from 2009 to 2011. Camera traps were active for an average of 87 (SD = 32) consecutive days between August 2009 and May 2010 in Temengor, and 82 (SD = 19) consecutive days

between August 2010 and April 2011 in Belum. The camera setup covered an area of approximately 400 km$^2$ in each habitat (*Rayan & Linkie, 2015*). A grid of 70 cameras was created in Belum and Temengor, respectively, with each grid cell covering 2 × 2 km. To increase spatial coverage, the cameras were moved within the grid after 3 to 4 months of operation, resulting in 140 distinct camera locations. The sampling blocks within Belum and Temengor were selected to represent the entire forest by taking into account the proportion of different vegetation types (*Rayan & Linkie, 2015*).

The placement of camera traps (RECONYX and SONY P41) in PFR and BTFC was chosen to ensure an average distance of about 1 km between traps (Fig. 1). As the approximate home range size of *M. nemestrina* is 1 km$^2$ (*Ruppert et al., 2018*; *Holzner et al., 2019*), a macaque group was unlikely to be detected by two different cameras, ensuring spatial independence between sites. Camera traps were active for 24 h per day and set to take photos at 10-s intervals. They were fixed to trees at a height of approximately 50 cm above the ground. Ground trapping (as opposed to placing cameras higher up in the trees) was reasonable for this species, as *M. nemestrina* has previously been described as a predominantly terrestrial primate, spending on average 56% of its active time on the forest ground (*Ruppert et al., 2018*). GPS locations of the cameras were recorded using Garmin GPSMAP® 60CSX hand-held GPS units.

The rationale behind including data from two different sites, *i.e.*, PFR and BTFC, despite methodological differences was to provide a broader perspective on how distinct forms of human disturbance may differentially affect macaque site occupancy, and thus to increase the analytical power of this study. While PFR represents a highly degraded habitat that is affected by partial clear-cutting, data from BTFC may specifically inform about the potential impact of selective logging. Further, the analysis of PFR was specifically focused on describing dynamics in pig-tailed macaque site occupancy over time, whereas the analysis of BTFC focused on providing thorough insight into spatial differences between undisturbed and selectively logged forests.

## Detection histories

Based on presence and absence data obtained from photographic records, we constructed detection histories for each camera site in PFR and BTFC. For repeated sampling occasions, we recorded a '1' when macaques were detected and a '0' when no macaques were detected even though a camera trap was active, either because they were truly absent from a particular site or because they were outside the detection range of a camera trap. Referring to previous studies (*Tan et al., 2017*; *Semper-Pascual et al., 2020*), we pooled daily detection/non-detection records for each camera site into sampling occasions of seven (PFR) and 14 (BTFC) consecutive days, respectively, in order to minimize the risk of temporal interdependence among occasions and to increase the overall detection probability. The periods of 1 and 2 weeks, respectively, were chosen to maximize the model fit according to the different data collection methods used across sites, as low probabilities of detection can prevent model convergence (*Dillon & Kelly, 2007*; *Tan et al., 2017*). As several camera traps within BTFC intermittently failed to record data for at least two sampling occasions and thus were excluded from analysis, the final datasets of Belum
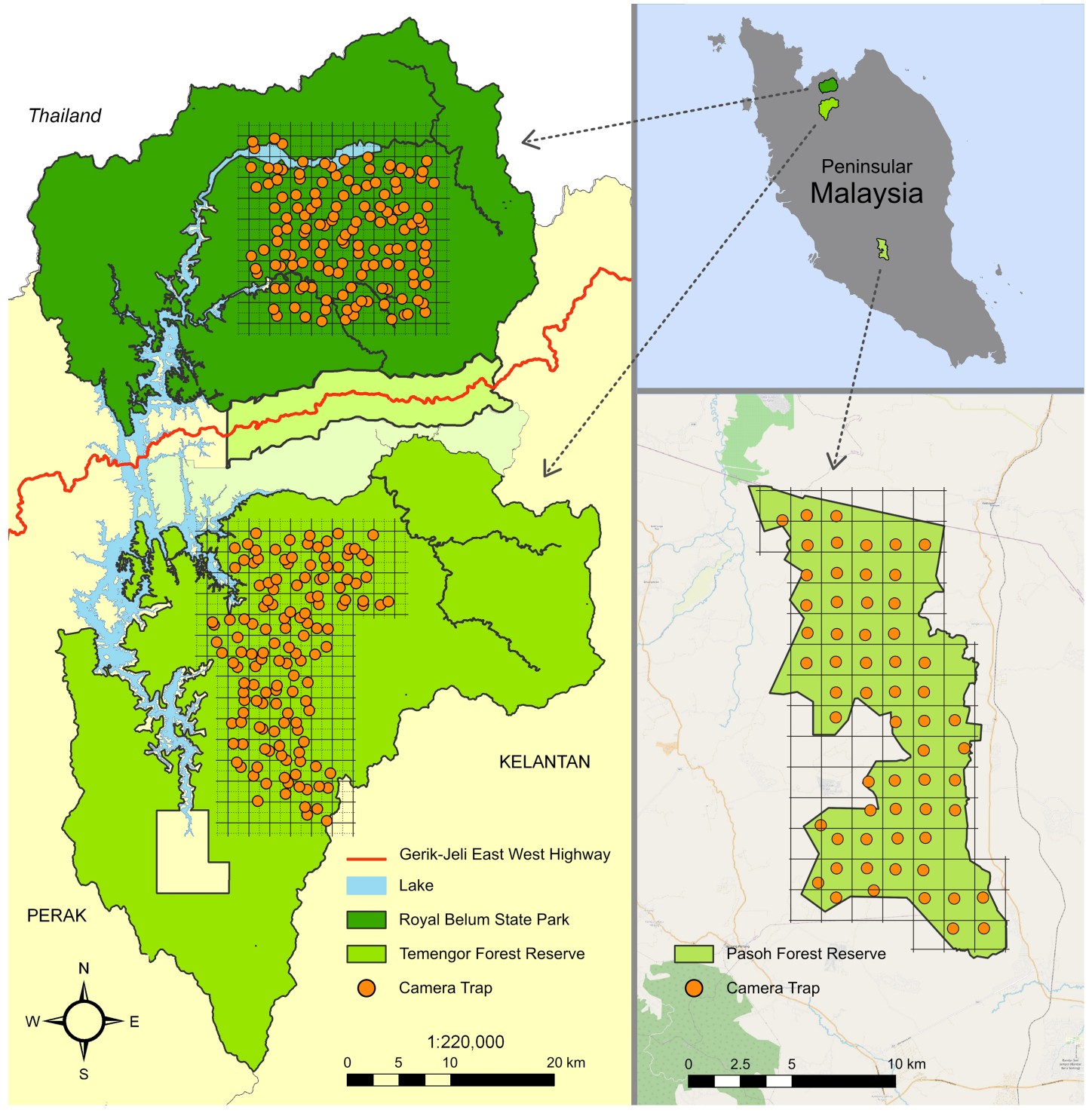

**Figure 1 Study sites in Peninsular Malaysia.** The maps indicate the camera trap distribution in the Belum-Temengor Forest Complex (left) and Pasoh Forest Reserve (right). Adapted from *Darmaraj (2012)*.

and Temengor included 129 and 125 camera sites, respectively. In PFR, all 60 camera traps were functioning. The total number of sampling occasions per year ranged between two and nine (mean ± SD = 5.5 ± 2.0) in PFR and between two and eleven (mean ± SD = 7.0 ± 1.9) in BTFC.

## Occupancy modelling

Using a maximum likelihood approach, occupancy models account for imperfect detection by linking a state model determining occupancy ($\psi$, $i.e.$, the probability with which a species occurs at a specific site) with an observation model determining detection probability ($p$, $i.e.$, the ability to detect a species when it is present) based on repeated samples from the same site (*MacKenzie et al., 2002*). Variation in probabilities across sampling sites and observation periods can be modelled using site-specific (*e.g.*, habitat variables) and observation-specific (*e.g.*, the sampling effort) covariates.

Firstly, to assess temporal changes in the macaques' distribution in the PFR, we fit the dynamic occupancy model described by *MacKenzie et al. (2003)*. This assesses site occupancy dynamics over multiple seasons by estimating, besides detection ($p$) and first year occupancy ($\psi$), the probabilities of colonization ($\gamma$, $i.e.$, the probability that a species is present at a previously unoccupied site) and extinction ($\varepsilon$, $i.e.$, the probability that a species is absent at a previously occupied site, *MacKenzie et al., 2003*). We included the mean elevation per camera grid cell (range = 94–664 m, mean ± SD = 295 ± 156 m) and the shortest distance to the forest edge (range = 5–2,865 m, mean ± SD = 1,076 ± 695 m) as potential predictors for first year $\psi$. Both altitude, as well as edge effects, which are likely to affect microclimate and tree communities in areas near the forest border, are known to be important factors in predicting the distribution of wildlife species, including primates (*McCain & Grytnes, 2010*; *Brodie, Giordano & Ambu, 2015*). Further, a shorter distance of a camera site from the forest edge may facilitate human access and thus increase the hunting pressure (*Milner-Gulland & Bennett, 2003*). Additionally, we modelled variation in $\gamma$ and $\varepsilon$ using a yearly site-specific categorial covariate describing differences in the occurrence and intensity of forest degradation through tree felling between camera sites and sampling years. Based on previous studies reporting a high variation in species' responses depending on the severity of human activity (*Gibson et al., 2011*; *Tobias, 2015*), we distinguished between clear-cutting, $i.e.$, land use change resulting in the loss of the entire forest cover in a specific area, selective logging, $i.e.$, the removal of a limited number of economically valuable trees for the timber industry, and no disturbance/tree felling. As previous research highlighted that primate populations may still be critically affected several years after tree felling took place (*Shelton, 1985*; *Rao & Schaik, 1997*), we classified a site as clear-cut or selectively logged if the respective type of forest degradation occurred during the past 5 years. Finally, we included the survey effort, $i.e.$, the number of days a camera trap was active during a sampling occasion (range = 1–7, mean ± SD = 6.0 ± 1.8 days), the sampling month (Jan, Feb, May–Dec), and the sampling year (2013–2017) as predictors for $p$ in order to account for seasonal variation and the effects of abiotic factors on the macaques' activity (*Takemoto, 2004*;

*Hanya et al., 2018*). Based on the six predictor variables, we constructed the full model and subsequently built candidate models with all possible combinations of predictor sets.

Secondly, to assess the macaques' spatial distribution within the BTFC, we fit a single-season occupancy model (*MacKenzie et al., 2002*). We are confident that our dataset, including sampling periods of a maximum length of 5 months, meets the closure assumption of occupancy modelling (*i.e.*, that the occupancy status of a site does not change during the sampling, *MacKenzie et al., 2002*), as previous studies suggested home range areas of *M. nemestrina* to be stable even over several years (*Ruppert et al., 2018*; *Holzner et al., 2019*). As above, we included the mean elevation per grid cell (range = 323–1,664 m, mean ± SD = 737 ± 302 m) as a potential predictor for $\psi$. To account for the effects of habitat degradation on $\psi$, we further modelled the habitat type (undisturbed Belum or selectively logged Temengor) and the mean Normalized Difference Vegetation Index (NDVI, range = 0.45–0.85, mean ± SD = 0.78 ± 0.05). The NDVI estimates the density of green on a patch of land by measuring differences between visible and near-infrared reflectance of vegetation cover, thus having frequently been used as a proxy to describe the bio-structural changes in vegetation caused by land clearing and logging (*Weier & Herring, 2000*; *Rayan & Linkie, 2015*; *Hamel et al., 2009*). It was computed using ASTER satellite images from 2008 to 2011 with a 15-m spatial resolution (*Rayan & Linkie, 2015*). Additionally, we included the distance to the nearest human settlement, including indigenous villages and logging camps in the forest (range = 662–20,643 m, mean ± SD = 8,099 ± 4,756 m), as this may be indicative of the intensity of human hunting of macaques at a camera site (*Milner-Gulland & Bennett, 2003*). Unlike in PFR, we did not expect pronounced edge effects in BTFC, as it comprises huge areas of continuous forest habitat, with the majority of camera traps being located several kilometres from the forest border. The approximate home range area of *M. nemestrina*, on the other hand, is only 1 km$^2$. Finally, to account for variation in $p$, we included the survey effort (range = 1–14 days, mean ± SD = 12.1 ± 3.6 days) and the sampling month (Jan–May, Aug–Dec) as observation-specific predictors into the model. As described above, we constructed the full model based on all six predictor variables and then built candidate models with all possible combinations of these predictors.

We fitted the occupancy models for PFR and BTFC, respectively, using the functions *colext* and *occu* from the package 'unmarked' (version 1.0.1, *Fiske & Chandler, 2011*) in R (version 3.4.4, *R Core Team, 2018*). To facilitate model interpretation and convergence, we standardized all continuous predictors before model fitting to a mean of zero and a standard deviation of one (*Schielzeth, 2010*). We checked for collinearity between continuous covariates using Spearman's rank correlation. Covariates were considered independent if their correlation coefficient $|r_s| \leq 0.7$ (*Dormann et al., 2013*). This was the case for all covariate pairs included in the same model. We drew inference using multi model inference based on Akaike's information criterion (AIC, *Burnham & Anderson, 2002*). Following recommendations of *MacKenzie (2006)*, we first modelled detection, identifying a suitable covariate structure for $p$ while holding $\psi$ and, in the dynamic model of PFR, $\gamma$ and $\varepsilon$ at the most general model including all covariates. Having identified the most parsimonious model structure for $p$, we kept this constant and
modelled occupancy, colonization, and extinction, respectively. This two-step approach may be advantageous over maintaining a general model for $p$, as it reduces the number of parameters being estimated (*MacKenzie, 2006*). We assessed the role of our covariates on $\psi$, $\gamma$ and $\varepsilon$ by ranking all candidate models according to their AIC corrected for small sample sizes (AIC$_c$, *Burnham & Anderson, 2002*). We considered top-ranked models as those with $\Delta$AIC$_c \leq 2$ (*Burnham & Anderson, 2002*). Model estimates were obtained by averaging over all candidate models using the zero method (*Nakagawa & Hauber, 2011*). We tested the goodness of fit of the global models by comparing the observed Chi-square statistics to respective reference distributions calculated from 1,000 parametric bootstraps (*MacKenzie & Bailey, 2004*). This indicated no lack of fit for both models (both $P > 0.05$, further details and R functions used in Supplemental Methods).

**Assessment of the macaques' age and sex structure**

To gain a deeper insight into the viability of *M. nemestrina* in selectively logged forests, we compared age and sex ratios in the macaque populations in Belum and Temengor, respectively, based on camera trap images. Due to a combination of time constraints for completing analyses, and the fact that images were not immediately accessible from PFR, which would have required time to individually download, organise and review before age and sex classifications could be established, this assessment was restricted to BTFC.

Each individual detected was identified as adult male, adult female, subadult, juvenile, or infant according to its body size, sexual characteristics (*e.g.*, anogenital swelling and elongated nipples in females or prominent testes in males, *Bullock, Paris & Goy, 1972*), or individual behaviour (*e.g.*, juveniles ranging in frequent proximity to their mother or infants nipple holding). Individuals that were partly hidden from view and thus could not be clearly assigned to either of these categories were marked as 'unknown'. We then summed the number of independent detections in each age and sex class, separately for each of the two habitats. Following *O'Brien, Kinnaird & Wibisono (2003)* and *Kafley et al. (2019)*, we defined independent detections as (1) consecutive photographs of identifiable different individuals based on their unique characteristics, (2) consecutive photographs of individually unrecognizable macaques of the same age and sex class taken more than 30 min apart, or (3) non-consecutive photographs of individuals of the same age and sex class. As described above, spatial independence between camera sites was assumed due to the generally small home range size of approximately 1 km$^2$ of *M. nemestrina* (*Ruppert et al., 2018*; *Holzner et al., 2019*). The identification of individuals in images across cameras was not possible in the framework of this study. We assessed differences in the macaques' age and sex structure between Belum and Temengor using a Chi-square test for independence.

## RESULTS

**Detection of macaques in PFR and BTFC**

Within PFR, we detected *M. nemestrina* during 42.3% of in total 1,636 independent sampling occasions. The naïve occupancy, *i.e.*, the proportion of camera sites with at least

**Table 1 Top-ranked *Macaca nemestrina* detection models (ΔAIC_c ≤ 2) for Pasoh Forest Reserve (PFR) and Belum-Temengor Forest Complex (BTFC) with global occupancy models.** Shown are Akaike's Information Criterion corrected for small samples (AIC_c), differences in AIC_c between each model and the respective best model (ΔAIC_c), the probability of each model to the best model, *i.e.*, the Akaike weights (wAIC), and the number of parameters (K, details on model selection and model averaged estimates for all covariates in Tables S1 and S2).

| Site | Top-ranked models | AIC_c | ΔAIC_c | wAIC | K |
|------|------|------|------|------|------|
| PFR | $p$ (effort + sampling year ) | 2,069.5 | 0 | 1 | 15 |
| BTFC | $p$ (effort + sampling month) | 1,238.6 | 0 | 0.995 | 16 |

one detection (MacKenzie et al., 2006), ranged between 0.80 and 0.93 during the 5-year sampling period, with the highest rate recorded in 2013. In the BTFC, macaques were present during 13.3% of 1,774 sampling occasions. The naïve occupancy was 0.53 in the undisturbed forest of Belum and 0.39 in the selectively logged forest of Temengor.

Based on AIC_c, camera trapping effort and the sampling date significantly contributed to explaining the variation in the detection probability of *M. nemestrina* (Table 1). Specifically, detection was positively correlated with the number of trapping days at both study sites, *i.e.*, PFR and BTFC (model estimate ± standard error (PFR/BTFC) = 0.51 ± 0.06/0.61 ± 0.12), and varied between sampling year and sampling month, respectively, indicating the presence of seasonal effects (details in Table S2). The mean estimated detection probability across all camera sites was 0.48 (SD = 0.15) in PFR and 0.23 (SD = 0.11) in BTFC.

## Temporal changes in macaque site occupancy in the highly disturbed PFR

Using a dynamic occupancy modelling approach, we aimed at predicting temporal changes in macaque site occupancy as well as their potential causes in PFR. The initial occupancy probability of *M. nemestrina* in PFR was estimated to be 0.95 (standard error (SE) = 0.03), when fixing elevation and the distance to the forest edge at their mean values. Macaque site occupancy in subsequent years was found to decrease by 10% from 0.95 in 2013 to 0.85 in 2017 (Fig. 2).

Further, we assessed the role of environmental and anthropogenic factors in predicting occupancy, colonization and extinction in PFR. Accordingly, only the top-ranked model, including forest degradation (arising from tree felling through clear-cutting or selective logging) as a predictor for extinction, received substantial support (ΔAIC_c ≤ 2, Table 2). In line with this, model-averaged coefficients corroborated the effect of forest degradation on the local extinction probability of *M. nemestrina* in PFR (Table 3). Specifically, we found that the macaques were approximately 6 times more likely to be absent at a previously occupied camera site in areas affected by clear-cutting compared to undisturbed forest patches (Fig. 3). No significant effect could be found for selective logging, yet large confidence intervals indicate a high variability in the response of *M. nemestrina* to this less intensive form of habitat degradation, potentially relating to small sample sizes as well as different practices and intensities of selective timber

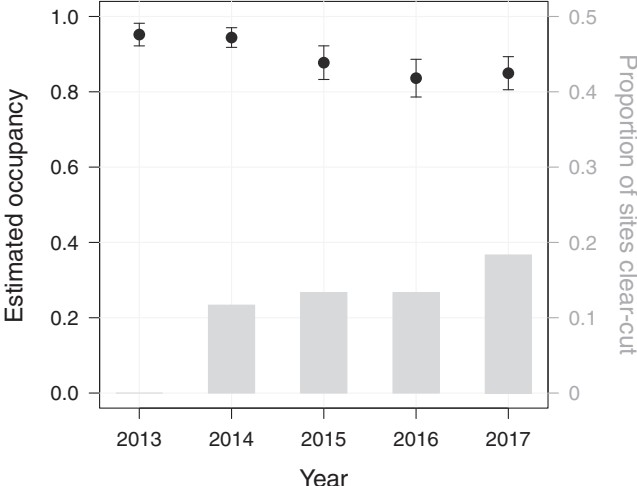

**Figure 2 Dynamics in *Macaca nemestrina* occupancy in the Pasoh Forest Reserve from 2013 to 2017.** Shown are site occupancy estimates, *i.e.*, the predicted proportion of sampled sites that are occupied, and their standard errors ($N = 60$). The bars indicate the cumulative proportion of sites at which clear-cutting occurred.

**Table 2 Top-ranked *Macaca nemestrina* occupancy models ($\Delta AIC_c \leq 2$) for Pasoh Forest Reserve (PFR) and Belum-Temengor Forest Complex (BTFC) with best respective detection models.** Shown are Akaike's Information Criterion corrected for small samples ($AIC_c$), differences in $AIC_c$ between each model and the respective best model ($\Delta AIC_c$), the probability of each model to the best model, *i.e.*, the Akaike weights (wAIC), and the number of parameters (K, details on model selection in Table S3).

| Site | Top-ranked models | $AIC_c$ | $\Delta AIC_c$ | wAIC | K |
|------|-------------------|---------|---------------|------|---|
| *PFR* | $\psi$ (.) $\gamma$ (.) $\varepsilon$ (forest degradation) | 2,056.4 | 0 | 0.633 | 11 |
| *BTFR* | $\psi$ (elevation) | 1,232.1 | 0 | 0.415 | 13 |

harvesting (Fig. 3). Further, low model-averaged coefficients and 95% confidence intervals including zero suggest that both elevation and the distance of a camera site to the forest edge did not significantly affect macaque site occupancy in PFR (Table 3).

## Macaques' spatial distribution in undisturbed and selectively logged forests within BTFC

To better understand the effects of selective logging on *M. nemestrina*, we assessed the macaque distribution as well as the covariate structure that best explains variation in occupancy probabilities within BTFC. Unlike in PFR, we found evidence that elevation had a strong effect on site occupancy in BTFC, as indicated by its inclusion in the top-ranked model ($\Delta AIC_c \leq 2$, Table 2). Specifically, occupancy probability was found to significantly decrease with increasing elevation (Table 3, Fig. 4). Further, low model-averaged coefficients and comparatively large 95% confidence intervals suggest that occupancy did not significantly differ between habitats. Predicted occupancy probabilities in the strictly protected forest of Belum and the selectively logged forest of Temengor were 0.59 (SE = 0.08) and 0.58 (SE = 0.09), respectively, when fixing all other covariates at

**Table 3 Effect of covariates on *Macaca nemestrina* occupancy, colonization and extinction in the Pasoh Forest Reserve (PFR) and Belum-Temengor Forest Complex (BTFC).** Shown are model averaged estimates (zero method), standard errors (SE) and lower and upper 95% confidence intervals (CI). Predictors included into the respective top models ($\Delta AIC_c \leq 2$) are indicated in bold.

| Site | Parameter | Covariate | Estimate | SE | lower CI | upper CI |
|------|-----------|-----------|----------|-----|----------|----------|
| *PFR* | Occupancy $\psi$ | distance to edge[a] | −0.03 | 0.29 | −1.47 | 1.14 |
| | | elevation[a] | −0.04 | 0.29 | −1.51 | 1.12 |
| | Colonization $\gamma$ | forest degradation (no *vs.* clear-cut)[b] | 0.02 | 0.33 | −2.58 | 3.58 |
| | | forest degradation (no *vs.* selective)[b] | −0.01 | 0.34 | −3.52 | 2.81 |
| | Extinction $\varepsilon$ | **forest degradation (no *vs.* clear-cut)[b]** | **2.25** | **0.69** | **1.13** | **3.48** |
| | | **forest degradation (no *vs.* selective)[b]** | **0.60** | **1.06** | **−1.47** | **2.70** |
| *BTFC* | Occupancy $\psi$ | habitat (Belum = 0, Temengor = 1) | −0.001 | 0.23 | −0.90 | 0.90 |
| | | NDVI[a] | 0.02 | 0.10 | −0.28 | 0.45 |
| | | distance to settlement[a] | −0.01 | 0.11 | −0.46 | 0.35 |
| | | **elevation[a]** | **−1.17** | **0.23** | **−1.62** | **−0.73** |

Notes:
[a] z-transformed to mean = 0 and SD = 1 prior to model fitting; original means ± SDs were: distance to edge: 1,076 ± 695 m, elevation (*PFR*): 295 ± 156 m, NDVI: 0.78 ± 0.05, distance to settlement: 8,099 ± 4,756 m, elevation (*BTFC*): 737 ± 302 m.
[b] Reference level is 'no tree felling'.

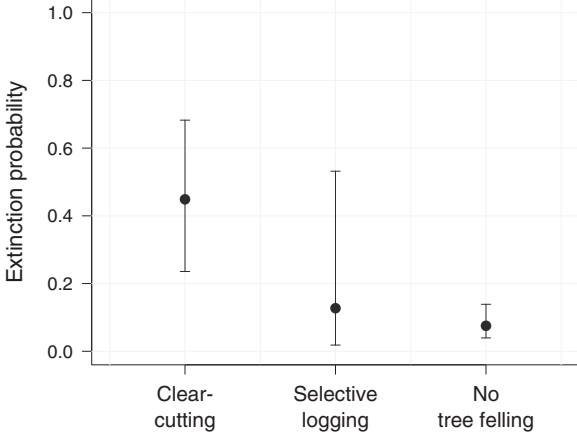

**Figure 3 Effect of forest degradation on the local extinction probability of *Macaca nemestrina* in the Pasoh Forest Reserve.** The filled circles show the fitted model and the whiskers its 95% confidence interval, conditional on all other predictors being fixed at their mean values. Clear-cutting occurred at a total of 11 sites during the study period, while five sites were selectively logged and 44 sites remained undisturbed (*N* = 60).

their mean values. Similarly, the NDVI and distance to the closest human settlement had no effect on macaque site occupancy (Table 3).

## Macaques' age and sex structure in BTFC

To explicitly examine the viability of *M. nemestrina* in selectively logged forests, we investigated whether the macaques' age and sex structure differed between intact and partially degraded habitats within BTFC. We detected a total of 614 and 695 individual macaques in Belum and Temengor, respectively, 96% of which could be unambiguously

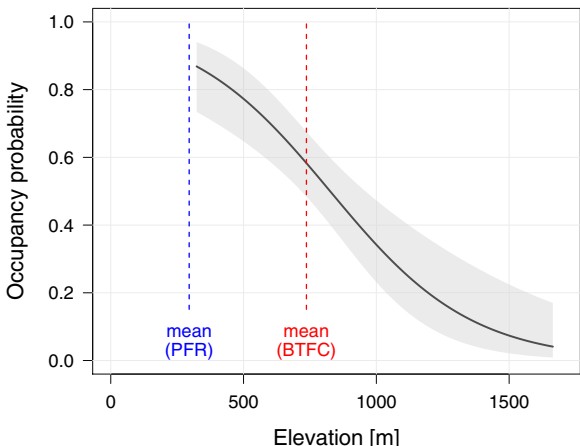

**Figure 4** **Effect of elevation on site occupancy of *Macaca nemestrina* in the Belum-Temengor Forest Complex.** The solid line shows the fitted model and the shaded areas its 95% confidence interval, conditional on all other predictors being fixed at their mean values (N = 254). The dashed lines indicate the mean elevation at the two study sites, *i.e.*, the Belum-Temengor Forest Complex (BTFC) and Pasoh Forest Reserve (PFR).

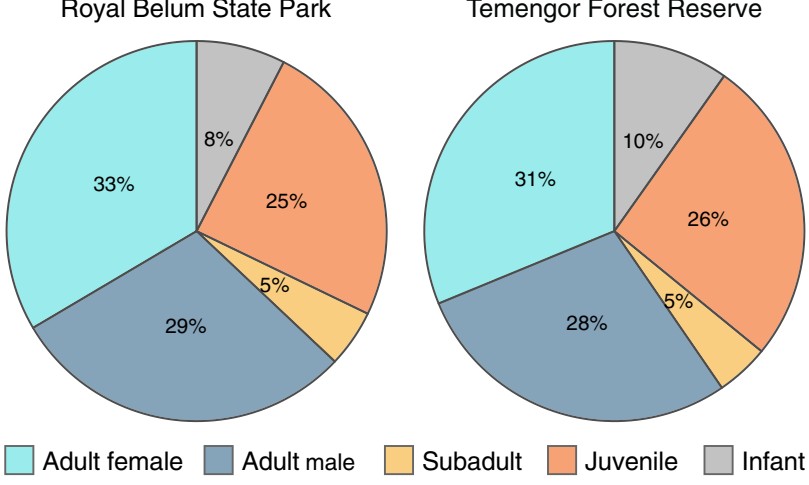

**Figure 5** **Age-sex structure of *Macaca nemestrina* in the Belum-Temengor Forest Complex.** The pie charts indicate the proportion of independent detections of each age-sex category, separately for the Royal Belum State Park (Belum, N = 594) and Temengor Forest Reserve (Temengor, N = 663).

assigned to an age and sex class. Chi-square test of independence did not reveal significant differences in the age and sex ratio between the undisturbed forest of Belum and the selectively logged forest of Temengor ($\chi^2$ = 0.45, df = 4, $p$ = 0.98). In both habitats, approximately 60% of detected individuals were adults, while 40% of detections were immatures including subadults, juveniles and infants (Fig. 5).

## DISCUSSION

Globally, increasing human encroachment into natural habitats is a major cause of biodiversity loss (*Marques et al., 2019*), yet previous studies have highlighted the high

variability in species' ability to cope with anthropogenic impact (*Brodie, Giordano & Ambu, 2015*). This study provides the first insights into the distribution of *M. nemestrina* across intact and degraded forest habitats in Peninsular Malaysia, focusing on the effects of tree felling on macaque occupancy over time and space. Specifically, clear-cutting significantly increased the probability that *M. nemestrina* became locally extinct at a previously occupied site as compared to a site without tree felling, likely accounting for the considerable decline in site occupancy observed in PFR during the 5-year study period from 2013 to 2017. However, there was no difference in the extinction probability of *M. nemestrina* between selectively logged and undisturbed sites within PFR, which is consistent with our findings from BTFC where occupancy probabilities did not depend on whether a site was located in the strictly protected forest of Belum or the selectively logged forest of Temengor. Importantly, all occupancy measures were obtained while controlling for differences in the detection probability of *M. nemestrina* between the study sites. Finally, there were no differences in the macaques' age and sex structure between Belum and Temengor, suggesting that low to moderate habitat degradation, such as selective logging, is not necessarily linked to declining populations, as would be indicated by an increased ratio of adults to immatures (*Rudran & Fernandez-Duque, 2003*; *Shil, Biswas & Kumara, 2020*). Rather, this species may be able to maintain viable populations in selectively logged forests. Further, similar sex ratios do not imply negative effects of selective timber harvesting on the survival of dispersing males (*Rudran & Fernandez-Duque, 2003*; *Zunino et al., 2007*; *Klass, Belle & Estrada, 2020*).

As one of the world's leading palm oil producers, Malaysia continues to be affected by deforestation, which has dramatic consequences for many tropical species that rely on primary rainforest (*Vijay et al., 2016*; *Estrada et al., 2017*). Our results provide evidence that southern pig-tailed macaques are particularly threatened by intensive forest clearance. Clear-cutting for the purpose of converting natural forest, whether undisturbed or previously selectively logged, into other land use forms, including agricultural land, is likely to dramatically reduce the suitability of a habitat for this species. In degraded forests, wildlife may suffer from habitat fragmentation, reduced availability of natural food sources, and a high human hunting pressure facilitated by the increased accessibility to the remaining forest patches (*Johns, 1985*; *Tilker et al., 2019*). Previous studies emphasized the primates' dependency on preserved natural forests in and around these highly disturbed landscapes to successfully disperse and reproduce (*Ancrenaz et al., 2021*), as well as to perform the full range of their natural behavioural repertoire (*Holzner et al., 2021*). The latter includes the formation of strong social relationships, which are critical characteristics of group-living animals owing to their crucial link to individual fitness (*Cameron, Setsaas & Linklater, 2009*; *Schülke et al., 2010*; *Ellis et al., 2019*) and offspring survival (*Silk, Alberts & Altmann, 2003*).

In the light of the rising global demand for palm oil, it is inevitable to focus conservation actions on reducing the negative environmental impacts of oil palm monocultures. Recent research has assessed the viability of potential palm oil substitutes, such as sunflower and coconut oil or single cell oil from yeast and microalgae (*Parsons, Raikova & Chuck, 2020*). Yet, due to its high-per hectare yield and unique lipid profile, palm oil still

outweighs the available alternatives (*Parsons, Raikova & Chuck, 2020*). Therefore, the promotion of sustainable practices in the palm oil sector, including the avoidance of further deforestation, the refrainment from the use of chemical fertilizer and pesticides, as well as the establishment of green corridors and buffer zones, is of utmost importance to allow animals to pertain and survive in the forest-plantation matrix, and thus to prevent further loss of wildlife biodiversity.

Unlike clear-cut habitats, less intensively disturbed, selectively logged forests, may indeed sustain viable macaque populations under certain conditions. In this context, elevation in particular appears to be an important factor in predicting whether macaques occur at a given site. This is unsurprising, as elevation defines different floristic zones and thus determines food availability for a variety of species, such as the predominantly frugivorous southern pig-tailed macaque (*Saw, 2010*). Previous studies have highlighted the impact of elevation on the occurrence and abundance of wildlife. *McCain & Grytnes (2010)*, for example, found a general trend of declining species richness with increasing elevation across multiple taxa, including small mammals, reptiles, and amphibians. Further, *Campera et al. (2020)* reported a strong negative correlation between lemur abundance and elevation in the Malagasy rainforests. As predicted, we found macaque site occupancy to decrease with increasing elevation in BTFC. However, PFR lacks this correlation, likely due to low variation between camera sites and generally low altitudes not exceeding 670 m (range = 94–664 m). In BTFC, on the other hand, altitudes reached up to 1,600 m (range = 323–1,664 m). Importantly, this difference in altitude between the study sites may explain the general discrepancy between occupancy estimates in PFR and BTFC. While large parts of the BTFC comprise hill and upper dipterocarp forest of mid altitude as well as montane forest (*Rayan & Linkie, 2016*), PFR is a lowland rainforest (*Fletcher et al., 2012*), which was previously reported to be the preferred habitat type of *M. nemestrina* (*Yanuar et al., 2009*). This is in line with findings by *Goodman & Ganzhorn (2004)* who suggested that the average elevation used by primates in Asia is around 400 m.

Another important determinant of the ability of a species to occupy and persist in a habitat is the intensity of human activity, such as the hunting pressure. As demonstrated by *Tilker et al. (2019)*, intensive hunting by humans may be an even more immediate threat to tropical wildlife than moderate habitat degradation. Both the distance to human settlements and the distance to the forest edge were not included in our top-ranked occupancy models, indicating that hunting activities by indigenous tribes, local communities, and logging workers may have been rather low at our study sites. In BTFC, this may be closely linked to low densities of settlements, which entail greater distances averaging 8 km to camera sites.

Earlier studies suggested that species characterized by a more generalist diet, and thus a lower degree of frugivory, may thrive in partially logged habitats (*Johns & Skorupa, 1987*; *Vetter et al., 2011*). Some of these were found to even prefer disturbed environments to primary forest. Ungulates, small mammals or omnivorous and granivorous birds, for example, exhibit higher abundances in disturbed or edge-affected habitats compared to undisturbed forests (*Lambert, Malcolm & Zimmerman, 2006*; *Brodie, Giordano &*

*Ambu, 2015*; *Burivalova et al., 2015*). Although the main component of *M. nemestrina*'s natural diet are fruits (ca. 75%, *Caldecott, 1986*), they feed on a wide range of other foods, such as insects, leaves, mushrooms and small mammals (*Ang et al., 2020*). Southern pig-tailed macaques inhabiting a forest-oil palm matrix at the west coast of Peninsular Malaysia were reported to complement their natural forest diet with cultivated oil palm fruits and plantation rats (*Ruppert et al., 2018*; *Holzner et al., 2019*), suggesting that macaques may indeed be able to adapt their diet to changing environmental conditions, as also found for other Malaysian primates (*Johns, 1985*).

Previous findings stressed the importance of accounting for imperfect detection during data collection when studying the occurrence or distribution of wild animals (*MacKenzie, 2006*). This proved to be relevant also in our study, as the detection probability, *i.e.*, the probability to detect a species when it is present, varied across study sites. Camera sites in BTFC in particular showed a low probability to detect macaques when present, even after increasing the interval of sampling occasions from 7 to 14 days. In PFR, on the other hand, the probability of detection was considerably higher. One crucial factor in explaining this discrepancy in the detection of wildlife may be seasonality. Based on our results, the date of sampling was identified as an important predictor of the detection probability. Prolonged rainfall during the monsoon season may decrease the macaques' overall activity and/or terrestriality (*Takemoto, 2004*; *Hanya et al., 2018*), thus resulting in a lower probability of being detected by the camera traps on the ground. While camera trapping in BTFC was performed from August until May, including the rainy season from November until January, in PFR more than 87% of sampling days took place during the commonly dryer period between May and August, likely resulting in a higher detection probability in PFR compared to BTFC. This effect may be reinforced, as PFR is a relatively small, highly degraded forest surrounded by oil palm plantation, with canopy gaps likely promoting movement of macaques on the ground (*Ancrenaz et al., 2014*). As pointed out by previous research, small home range areas frequently reported for primate groups ranging in anthropogenic environments, as well as high group densities may also lead to increased detection probabilities (*McLennan, Spagnoletti & Hockings, 2017*; *Parsons et al., 2017*; *Neilson et al., 2018*), although the available dataset did not allow us to verify this. Furthermore, it is important to note here that, based on our analyses, we can infer macaque occupancy but not necessarily abundance. In order to provide in-depth information on whether or not selective logging affects the long-term viability of *M. nemestrina*, more detailed studies including larger data sets are needed. However, *MacKenzie & Nichols (2004)* proposed that occupancy may serve as a surrogate for abundance estimation and some earlier studies found strong associations between occupancy and density in carnivorous species (*Clare, Anderson & MacFarland, 2015*; *Linden et al., 2017*).

## CONCLUSIONS

There are no population assessments of *M. nemestrina* in its species range, but general estimates are primarily based on assumptions inferred from knowledge available from other primates occupying the same or similar habitats (*Ang et al., 2020*). Here, we add to

these findings by providing thorough insight into the macaques' ability to persist in human-impacted habitats and quantifying the effect of tree felling activities on the distribution of *M. nemestrina*. We confirm how population monitoring through camera trapping can contribute to understanding the response of an elusive and threatened Malaysian primate to ecological and anthropogenic factors, and hence to informing conservation efforts. The present study stresses the high sensitivity of *M. nemestrina* to clear-cutting. At the same time, it demonstrates that not only primary forest but also moderately disturbed habitats may play a key role for the protection of this species. Overall, our data indicate that previously selectively logged forests may constitute a valuable habitat for the macaques and therefore should be protected and regenerated instead of opened for more land development. Ultimately, it is imperative to clearly differentiate between these partially degraded, but for the protection of biodiversity, very important forests (*Johns, 1985*; *Lee, Powell & Lindsell, 2015*) and vast areas of monoculture timber plantations. Frequently, the latter are also defined as 'forest' (*e.g.*, Peninsular Malaysia's National Forestry Act of 1984) and thus continue to legally replace selectively logged areas, *i.e.*, potential primate habitats, in many forest reserves after the high-value forest timber had been extracted (*Aziz, Laurance & Clements, 2010*). To counteract population declines at accelerated rates, conservation actions need to focus on the maintenance (and if necessary, restoration) of primary and secondary forest habitats (*WWF, 2020*), including partially degraded forest that can provide valuable habitat for various species, such as *M. nemestrina*. Specifically, the protection of selectively logged forest against conversion into other land use forms, *e.g.*, monoculture plantations, targeted restoration efforts of degraded habitats, and the reconnection of isolated forests through the establishment of wildlife corridors in fragmented habitats are important conservation measures. This may facilitate natural dispersal between wildlife populations, which is inevitable to ensure the long-term survival of this and other species.

## ACKNOWLEDGEMENTS

We thank the Perak State Parks Corporation, the Department of Wildlife and National Parks and the Forestry Department of Perak for permission to conduct surveys. We are very grateful to the Tropical Ecological Assessment and Monitoring (TEAM) Network and the WWF Malaysia for sharing camera trap data on southern pig-tailed macaques. A full account of all parties to be credited for the collection of the original data and research funding is given in *Rayan & Linkie (2015*, *2016*, *2020)* and *Tan et al. (2017)*. We also thank Monika Sündermann and Brigitte Schlögl for their support in the initial phase of this study, and Ammie Kalan for her advice on the application of occupancy modelling.

### Funding

This study was supported by the Ministry of Higher Education Malaysia for Fundamental Research Grant Scheme with Project Code: FRGS/1/2018/WAB13/USM/02/1 (awarded to

N.R.), University of Leipzig ('Doktorandenförderplatz' #G00042), the German Academic Exchange Service (DAAD) and the German Society of Primatology (GfP, all to A.H.). WWF-Malaysia's camera trapping in the Belum-Temengor Forest Complex was supported by WWF-Netherlands, the U.S. Fish and Wildlife Service, the Mohamed bin Zayed Species Conservation Fund and the Malaysian Wildlife Conservation Fund (all to D.M.R). The funders had no role in study design, data collection and analysis, decision to publish, or preparation of the manuscript.

## Grant Disclosures

The following grant information was disclosed by the authors:
Ministry of Higher Education Malaysia for Fundamental Research Grant Scheme with Project Code: FRGS/1/2018/WAB13/USM/02/1 (awarded to N.R.).
University of Leipzig ('Doktorandenförderplatz' #G00042).
German Academic Exchange Service (DAAD).
German Society of Primatology (GfP, all to A.H.).
WWF-Malaysia's.
Belum-Temengor Forest Complex.
WWF-Netherlands.
U.S. Fish.
Wildlife Service.
Mohamed bin Zayed Species Conservation Fund and the Malaysian Wildlife Conservation Fund (all to D.M.R).

## Competing Interests

Anja Widdig is an Academic Editor for PeerJ.

## Author Contributions

- Anna Holzner conceived and designed the experiments, analyzed the data, prepared figures and/or tables, authored or reviewed drafts of the paper, and approved the final draft.
- D. Mark Rayan performed the experiments, authored or reviewed drafts of the paper, and approved the final draft.
- Jonathan Moore performed the experiments, authored or reviewed drafts of the paper, and approved the final draft.
- Cedric Kai Wei Tan performed the experiments, authored or reviewed drafts of the paper, and approved the final draft.
- Laura Clart conceived and designed the experiments, analyzed the data, authored or reviewed drafts of the paper, and approved the final draft.
- Lars Kulik analyzed the data, authored or reviewed drafts of the paper, and approved the final draft.
- Hjalmar Kühl conceived and designed the experiments, authored or reviewed drafts of the paper, and approved the final draft.

- Nadine Ruppert conceived and designed the experiments, authored or reviewed drafts of the paper, and approved the final draft.
- Anja Widdig conceived and designed the experiments, authored or reviewed drafts of the paper, and approved the final draft.

### Animal Ethics

The following information was supplied relating to ethical approvals (*i.e.*, approving body and any reference numbers):

Due to the non-invasive nature of the data collection through camera trapping, which formed the basis of the present study, no ethical approval was required.

### Field Study Permissions

The following information was supplied relating to field study approvals (*i.e.*, approving body and any reference numbers):

For Temengor Forest Reserve, the forest entry permit was given to WWF Malaysia by the district Hulu Perak Forestry Department (through the Perak State Forestry Department), and for Royal Belum State Park, this was given by the Perak State Parks Corporation in 2007.

In addition, for the camera trapping research in Belum-Temengor in 2011, WWF-Malaysia had clarified on the need for a wildlife research permit and received the response from the Department of Wildlife and National Parks (DWNP) stating that they had no objection to conduct a noninvasive wildlife study if it is carried out in areas not governed by DWNP.

### Data Availability

Full datasets cannot be shared publicly because they contain location data of protected mammal species of Peninsular Malaysia. Basic detection data are available in the Supplemental Files. Data from Belum-Temengor are co-owned by WWF-Malaysia and one of the co-authors (D. Mark Rayan) who, at the time of data collection, was formally affiliated with WWF-Malaysia. Upon reasonable request, these data can be made available through an alternative contact from WWF-Malaysia (Christopher Wong, christopher.wong@wwf.org.my). Data from Pasoh were obtained as part of the Tropical Ecological Assessment and Monitoring (TEAM) Network work, conducted in collaboration with the Forest Research Institute Malaysia (FRIM), and can be made available upon reasonable request through one of the co-authors (Jonathan Moore, jonathan.moore03@gmail.com). Statistical analyses were done using the software R (version 3.4.4). R scripts are available in the Supplemental Files.

### Supplemental Information

Supplemental information for this article can be found online at http://dx.doi.org/10.7717/peerj.12462#supplemental-information.

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
