# Peer review of "Occupancy of wild southern pig-tailed macaques in intact and degraded forests in Peninsular Malaysia"

_PeerJ, doi:10.7717/peerj.12462_

## Round 0.1 · original submission · Minor Revisions

Overview
This manuscript examines the relationship between the occurrence of southern pig-tailed macaques and forestry practices using data from two earlier camera trap studies in Malaysia. At one site, average occupancy decreased over a 4-yr period, and the decrease was greater at clear-cut locations than at undisturbed or selectively logged locations. At the other site, there was no difference in occupancy between an undisturbed and a selectively logged location. At neither site was there significant evidence of an effect of distance to a human settlement, but the second site showed an effect of elevation. Age and sex classes were also similar between the undisturbed and selectively logged area.

Overall, this is a well written manuscript with a clear organization and relatively few grammatical errors. Both reviewers agree that it makes a valid scientific contribution and have provided a number of useful suggestions to clarify its message. Because their suggestions are primarily focused on revising the text for clarity, I consider this to be minor revisions, even though both reviewers listed them as major.

Editor’s Specific Comments

I agree with Reviewer 1 that the paragraph introducing the study species (L62ff) would fit better after more general material (after the paragraph ending on L108).

I also agree that the topic of demography in relation to habitat disturbance should be introduced.

Reviewer 1 asks you to add hypotheses. If you had hypotheses at the start of the study, it would be appropriate to mention them. However, if you did not have specific hypotheses, it would not be scientifically valid to add them now. A descriptive study to see if there are effects of two levels of logging disturbance without hypotheses is acceptable.

Reviewer 2 points out that the raw data have not been provided as required.

I agree with Reviewer 2 that a bit more of an effort to explain the relationship between the two studies would be helpful to readers. In relation to one of the comments by Reviewer 1, you might clarify to what the validity of comparisons between sites are relevant as compared to comparisons within sites.

I have provided a pdf with a number of minor grammatical suggestions and ways to remove redundant wording. These are indicated by highlights and inserted comments. After pointing out an issue, I did not necessarily highlight all subsequent cases. Please check carefully for other examples.

Congratulations on the Reference section. Unlike most manuscripts that I see, I did not notice any errors in formatting.

Reviewer 1 ·

Basic reporting

The authors present a generally sound paper, with strong methods and results sections. The figures and tables are of high quality, and the raw data are shared. The structure is now always coherent, please find more specific comments in the general comments section.

Experimental design

The authors used a clear experimental design that is well presented and with strong links to published literature, mainly from the same research group. Some of the methods are referred too much to previous papers, while some more information are required so that the reader do not have to find other papers to get important information. For example, I would add the resolution and the source of the raster you used to obtain NDVI in this paper. That is an important information that otherwise the reader would have to find in Rayan & Linkie, 2015. I suggest checking the methods to see if some of these important information that refer to previously published papers might be added.
The research aims are clearly stated, although they are descriptive and no predictions are presented. The authors should include specific research questions and hypotheses following the general structure of the journal.

Validity of the findings

The authors did a good job in presenting the data clearly, and I have no issues with their data analysis. What I found weak is the data interpretation, as the authors are missing to discuss some important implications of their findings. The structure of the discussion is also not always clear as the authors do not present clear research questions and hypotheses. The discussion is too specific on the studied species and context. The authors often discuss arguments that are not expanded in the discussion and introduced in the introduction. For example, line 371-373, you do not explain why you did that analysis, you do not introduce the topic, and you do not expand on your claim. That is valid also for other findings that you do discuss but you do not have in your introduction (you are mainly missing to introduce clear hypotheses and that is reflected in your discussion). Please find more specific comments in the general comments section.

Additional comments

I have two major concerns that I suggest to be solved before considered this paper for publication.
1) The introduction in not well structured. You need a funnel structure and not going into your specific context in the 4th line. It is now difficult to follow. I suggest restructuring your introduction having a more coherent narrative. For example, lines 74-108 are very broad and should be presented earlier. You do not have clear research questions and hypotheses.
2) I think the authors here are missing to expand the discussion on why they found higher probabilities of detection in the small forest adjacent to the oil palm plantations than in the relatively undisturbed forests. They just list some possible reasons (lines 174-180) but do not expand on that, and that weakens your paper. A reader may think that, if you do not have differences between selectively logged and undisturbed forests in BTFC, and the detection in PFC is even higher than in BTFC, the conclusion is that Macaca nemestrina prefers disturbed habitats adjacent to plantations. It is true that you found a decrease in the detection between 2013 and 2017 in PFC and that it is related to the increase in forest clearing, but the explanation why PFC has higher detection rates is not expanded. I suggest included some more clarification.
I have two arguments that I think should be expanded, but I guess the authors should think a bit more about expanding their current arguments as well as they now read very speculative and unsupported.
a) The first argument I would add is about elevation. PFR is more a lowland rainforest while the BTFC is more mid altitude. If you check Goodman and Ganzhorn (2004), they suggest that the average intermediate elevation point for primates in Asia is around 450 m a.s.l., and Macaca nemestina prefers lowland habitats (check also Yanuar et al. 2009). You can also check a work from Campera et al. (2020) who made an extensive argument on the importance of elevation on animal abundance.
Goodman SM, Ganzhorn JU (2004) Elevational ranges of lemurs in the humid forests of Madagascar. International Journal of Primatology 25: 331–350.
Yanuar A, Chivers DJ, Sugardjito J, Martyr DJ, Holden JT (2009) The Population Distribution of Pig-tailed Macaque (Macaca nemestrina) and Long-tailed Macaque (Macaca fascicularis) in West Central Sumatra, Indonesia. Asian Primates Journal 1: 2–11.
Campera, M., Santini, L., Balestri, M., Nekaris, K. A. I., & Donati, G. (2020). Elevation gradients of lemur abundance emphasise the importance of Madagascar’s lowland rainforest for the conservation of endemic taxa. Mammal Review, 1, 25–37.
b) The other argument I suggest to expand is that there are several species that actually prefer disturbed habitats, especially a matrix of forest and plantations. This argument is almost absent while that is very important as your data goes in that direction. Also, since you are inevitability discussing the impact of palm oil plantations, I suggest also to include discussion over this topic, integrating for example the following paper:
Parsons et al. (2020). The viability and desirability of replacing palm oil. Nature Sustainability

Reviewer 2 ·

Basic reporting

This paper in general is well-written, clear, and concise. However, there are some grammatical mistakes (primarily missing articles) throughout that should be addressed. I highlighted some of these in the general comments below.

Experimental design

It is not immediately clear from the abstract or the introduction why these two different data sets and analyses were included in this paper. Besides the use of camera traps on the same species, all other aspects (time period, sampling period, number of cameras, camera configuration, covariates, etc.) are different between the two sites making direct comparison difficult. It appears that the two sites differ in terms of the amount of degradation (though PFR is perhaps similar to TFR). Right now, the two analyses are very disjunct – making a clear story that connects the two and explains why they are both important for reaching the conclusions of the paper will really strengthen this study.
Generally, the knowledge gap and methods are clearly defined and detailed. However, I have noted a few comments below where extra clarification could be useful primarily in the methods and discussion.

Validity of the findings

This study is not currently replicable as covariate information is not included. It would be great to include this as well as the associated R code, if possible.

The conclusions are well-stated; however, there are a few additional questions highlighted in the general comments below that could warrant addressing or further discussion.

Additional comments

Here are some line-by-line comments:

L33: provide the first insights
L40: should this be no effect of selective logging on prob of ext? this variable wasn’t included on occupancy
L42-44: How does having the same age and sex structure suggest that selectively logged forest are important habitats?
L103: indicative of the ability for a species
L124: remove ‘totally’
L127: remove ‘about’
L157: Can standard deviation be added (as an error measure) on all of the averages for number of days that cameras are active?
L169-179: This information on camera trap settings and the reasoning behind why ground cameras are more appropriate than arboreal cameras for this species should be included in the paper instead of the figure legend. This could also be an interesting point to make in the discussion.
L188: why were different sampling occasion lengths used for each data set? What do you mean by ‘chosen to ensure a good model fit’?
L191: were all cameras functioning in PFR?
L201: remove ‘correctly’
L217: this section is unclear – what values can a site take for forest degradation – or is it a categorical covariate? If so, the category it falls within is based on what has happened in that site in the 5 years before camera trapping began? Does this category change over the study or each site is put into a category for the whole study?
Did colonization or extinction change over time? Was this tested? It seems that depending on when the clear cutting or selective logging happened this could affect these model parameters.
Were any of the covariates include in the same model correlated? It seems there could be a correlation between elevation and distance measures.
L247: why was distance to the nearest human settlement used instead of distance to the forest edge like in the analysis for PFR? It is not clear why different covariates were used for each analysis.
L271-272: this is results not methods
Why were different covariates used for each analyses – this makes comparison between the two study sites difficult.
L277: what is meant by limitations in data availability? TEAM data is all freely available.
L304: AICc?
L320: through?
L324: six-fold?
L339-340: this is discussion, not results – remove
L365: time and space?
L366: as compared to sites with no tree felling?
L368: this is confusing – selective logging is not tested as a covariate against extinction – the category of forest degradation was – so maybe instead you could say that there was no difference in extinction prob between selective logged sites and sites with no tree felling? (unless forest degradation was actually 3 separate covariates one for each category – but this is unclear in the methods)
L373: what are these certain conditions? It would be good to have more discussion about the age/sex structure.
L374: how are these values comparable when the sampling occasions are different? Even if the sites with fewer days is higher, doesn’t mean they are comparable.
L380: could this also be because ground cameras were used not arboreal cameras – if the species is likely to have increased movement on the ground in degraded forests?
L454: this study doesn’t address severe human encroachment – I would rephrase stresses the high sensitivity to clear cutting.
L455-460: delete this – this current study shows that areas that are selectively logged are also viable for this species

Figure 1: why do you the scale bars not start at 0? It seems in Royal Belum State Park and Temengor Forest Reserve some of the cameras are very close together – are these all more than 1 km apart? Does this spatial configuration of cameras, which differs from PFR, affect the estimates from the models?
Figure 2: What do you meant by smoothed site occupancy estimates? Aren’t these model averaged estimates?
Figure 3: Is it possible to show how many sites are selectively logged as well as clear cut? It seems that maybe there are few sites which is why there is such large error bars?
Table S2: Why are the estimates 0 for all the sampling months? For BTFC, it looks like sampling month Jan to Aug was not estimable?

---

## Round 0.2 · accepted · Accept

Thank you for your careful revision and clear response statement. I consider the manuscript now ready for publication. In my re-reading, I noticed only one minor error: L98 change to 'much of its forest is being converted' or 'many of its forests are being converted'. You should be able to make these corrections in proof.